# Digital exposure notification tools: A global landscape analysis

**Camille Nebeker**[1,2]**, Daniah Kareem**[1,2]**, Aidan Yong**[2]**, Rachel Kunowski**[3]**,
Mohsen Malekinejad**[4]**, Eliah Aronoff-Spencer** [2,5] *

**1** Herbert Wertheim School of Public Health and Human Longevity Science, University of California, San
Diego, California, United States of America, **2** The Design Lab, University of California San Diego, San Diego,
California, United States of America, **3** UC San Diego Health, San Diego, California, United States of
America, **4** California Department of Public Health, Sacramento, California, United States of America,
**5** Division of Infectious Diseases and Global Public Health, Department of Medicine, UC San Diego School of
Medicine, San Diego, California United States of America

* earonoffspencer@health.ucsd.edu

STATES

**Data Availability Statement:** All data are in the
manuscript and supporting information files.

**Funding:** The authors disclosed receipt of the
following financial support for the research,
authorship, and/or publication of this article: the

## Abstract

### Background

As the COVID-19 pandemic continues, digital exposure notification systems are increas-
ingly used to support traditional contact tracing and other preventive strategies. Likewise, a
plethora of COVID-19 mobile applications have emerged. *Objective*: To characterize the
global landscape of pandemic related mobile applications, including digital exposure notifi-
cation and contact tracing tools.

### Data Sources and Methods

The following queries were entered into the Google search engine: "(*country name*
COVID app) OR (COVID app *country name*) OR (COVID app *country name*+) OR
(*country name*+ COVID app)". The App Store, Google Play, and official government web-
sites were then accessed to collect descriptive data for each application. Descriptive data
were qualified and quantified using standard methods. COVID-19 Exposure Notification
Systems (ENS) and non-Exposure Notification products were categorized and summarized
to provide a global landscape review.

### Results

Our search resulted in a global count of 224 COVID-19 mobile applications, in 127 countries.
Of these 224 apps, 128 supported exposure notification, with 75 employing the Google
Apple Exposure Notification (GAEN) application programming interface (API). Of the 75
apps using the GAEN API, 15 apps were developed using Exposure Notification Express, a
GAEN turnkey solution. COVID-19 applications that did not include exposure notifications (n
= 96) focused on COVID-19 Self-Assessment (35·4%), COVID-19 Statistics and Information
(32·3%), and COVID-19 Health Advice (29·2%).

authors received funding from the California Department of Public Health, agreement 20-10777 (To UC San Diego Health, Project Sponsor Chris Longhurst; authors EAS, CN, RK received partial salary support for preparation). The funders had no role in study design, data collection and analysis, decision to publish, or preparation of the manuscript.

**Competing interests:** The authors have declared that no competing interests exist.

## Conclusions

The digital response to COVID-19 generated diverse and novel solutions to support non-pharmacologic public health interventions. More research is needed to evaluate the extent to which these services and strategies were useful in reducing viral transmission.

## Author summary

This study aimed to understand the global landscape of mobile applications related to the COVID-19 pandemic, particularly those focused on digital exposure notification and contact tracing. Data were generated by using specific Google search queries related to COVID-19 apps in different countries. Further information was obtained from the iOS App Store, Google Play, and official government websites, to collect information about each application. The study found a total of 224 COVID-19 mobile apps across 127 countries. Out of these, 128 apps supported exposure notification, with 75 of them utilizing the Google Apple Exposure Notification (GAEN) application programming interface (API). Among the GAEN API apps, 15 were developed using a turnkey solution called Exposure Notification Express. The remaining 96 apps did not include exposure notifications and focused on COVID-19 self-assessment, statistics and information, and health advice. Across all 224 applications, the authors found the digital response to COVID-19 resulted in a wide range of innovative solutions to support public health interventions. However, further research is needed to assess the effectiveness of these services and strategies in reducing the spread of the virus. In summary, this study provides an overview of the global landscape of COVID-19 mobile apps, highlighting the prevalence of exposure notification tools and other app categories. The findings suggest that these digital solutions have the potential to contribute to controlling the pandemic, but their actual impact needs to be evaluated through further research.

## Introduction

On March 11, 2020, COVID-19 was officially declared a global pandemic by the World Health Organization (WHO) [1]. By January 2022, over 304 million confirmed cases of COVID-19 and 5.4 million deaths were reported [2]. Efforts to reduce infection spread have led to myriad digital exposure notification tools to support traditional public health contact tracing strategies.

The goal of contact tracing is to identify those who have been exposed to a particular disease and isolate them before they further spread the disease. This disease mitigation method dates to the Middle Ages when red crosses were painted atop homes sick with the Plague and ships routinely quarantined for 40 days (Italian (venetian) quaranta giorni 'forty days') [3–5]. Traditional contact tracing works by identifying and contacting persons exposed to the infected individuals to see who they have been in contact with and so forth. Exposed contacts are notified and must isolate if they test positive, averting an otherwise new transmission chain. These methods have significant draw backs. Individuals may be reticent to engage with contact tracers, not be forthcoming with contacts, and may not remember their past contacts [6]. Further, Contact Tracing is challenging with a fast-spreading virus like COVID-19 as the virus can be transmitted without symptoms, making it time-consuming and hard to track using traditional contact tracing methods [7]. Today, digital solutions can supplement and work in a synergistic manner to address many of these shortcomings.

Digital Exposure Notification (EN) systems can play an important role in managing the spread of COVID-19 and other infectious diseases [8]. Digital exposure notification and traditional contact tracing have similar goals–that being to notify people who may have been exposed to the virus [9]. The ability to leverage technology to detect and communicate exposure can complement traditional methods and save lives [8]. By using technology to facilitate exposure notification, fewer human resources are needed, which can save money and time. When a person tests positive, they can anonymously alert people with whom they have been in close contact–including those who they may not even know they have exposed (e.g., person at the grocery store).

EN systems are designed to alert users of potential exposure to COVID-19, as well as to provide information and referral to testing and other forms of support. Bluetooth technology is one strategy used to support digital EN where anonymous "keys" are exchanged based on users' distance and duration of exposure. Some apps use Global Positioning System (GPS) tracking and QR code status as digital EN strategies, but most apps relied on Bluetooth technology for EN due to privacy protections [9,10]. In a partnership between Google and Apple, the Google Apple Exposure Notification System (GAEN) was developed in April 2020, which followed the PACT Protocol (Privacy Automated Contact Tracing) designed to preserve privacy in exposure detection methods [11]. GAEN was created for public health authorities (PHAs) to develop their own Exposure Notifications app using the GAEN API. Exposure Notifications Express (ENX) was a later development using GAEN. With ENX, PHAs provide Google and Apple with a configuration file that contains instructions, content, messaging, and risk parameters, which uses the GAEN API to generate a custom exposure notification express app [12].

The purpose of the current study was to document and characterize EN systems in place globally. By understanding where EN exists and related functionalities, we can characterize potentially beneficial strategies to reduce the spread of COVID-19.

## Methods

The purpose of this global landscape analysis was to document and characterize COVID-19 mobile applications, including Exposure Notification (EN) technologies.

### Eligibility criteria

The study eligibility criteria included any COVID-19 mobile application and COVID-19 Exposure Notification (EN) systems that were downloadable or activated on a mobile device. The search process involved using queries "(*country name* COVID app) OR (COVID app *country name*) OR (COVID app *country name*+) OR (*country name*+ COVID app)" in the Google search engine to identify COVID-19 apps in each country. The data included active and inactive apps discovered during the data collection period of June 2021—July 2021. The process by which the researchers distributed search replicated findings is described in Fig 1.

### COVID-19 app selection

The process for screening COVID-19 mobile apps involved first identifying if the COVID-19 app was a mobile app, which was defined as being able to download on a mobile device. Three co-authors (DK, AY, RK) divided a list of 195 countries into thirds. Each co-author then identified all COVID-19 apps that existed or previously existed in those countries. Our review includes both COVID-19 Exposure Notification (EN) apps and non-Exposure COVID-19 apps (non EN).

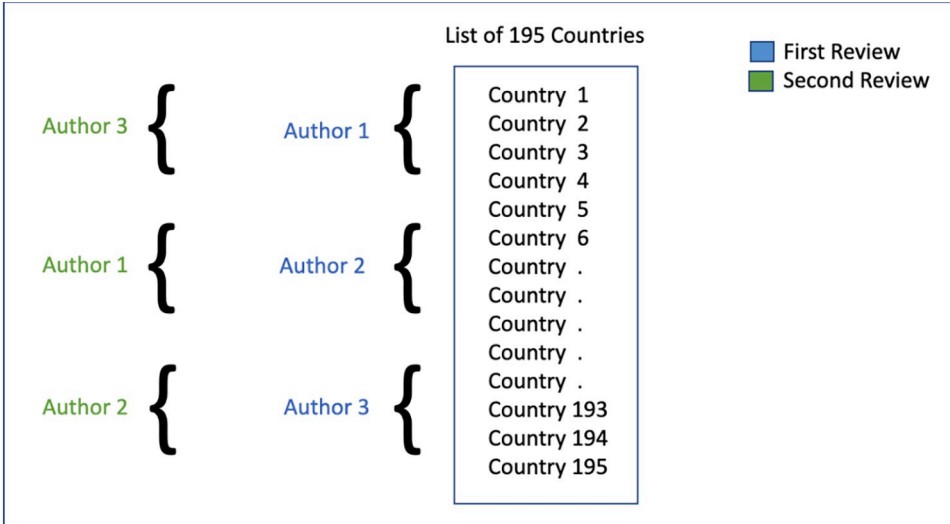

**Fig 1. Illustration of the distribution for search of COVID-19 apps by country and the review process between the three researchers in their search of COVID-19 apps.**

## Data collection process

Identified apps were entered into a database by country where the app was deployed along with descriptors of the app functionalities. When app descriptions in news articles, application websites, Google Play Store, or App Store were not available in English, Google Translate was used for translation.

The data collection process was divided into two phases. Phase 1 consisted of identifying every COVID-19 related mobile app using the above search method. For each resulting app, descriptive data were entered into a spreadsheet. The data included whether the: 1- app was an exposure notification app and, if yes, whether the API was GAEN, 2- product use was voluntary or mandatory in the country, and 3- product used privacy-first messaging on the official application website, Google Play Store, and/or App Store. In addition, the dataset included data management and design considerations.

Phase 2 involved separating apps into two categories: 1- Exposure Notifications apps (EN) and 2- Non-Exposure Notification apps (non EN). EN apps were further categorized as using GAEN and/or Exposure Notifications Express (ENX). For each COVID-19 app, the features were documented, and descriptive statistics applied.

Assumptions made in the process were that the information given on the Google Play store, App Store, and articles/websites discussing the application accurately described all app features. Table 1 below lists the app features and intended functionality.

An inter-rater reliability process was used to verify consistent coding during the data collection and analysis process. The app search process involved three researchers inputting the same keywords into the same search engine on three different laptops, (*country name* COVID App). The researchers checked for consistency in coding by searching for apps within the same 10 countries independently and then met to discuss the codes used to describe features and develop code definitions. Upon agreement of the coding schema, the search for global apps was divided among the three co-authors, with each responsible for searching countries within the assigned segment and documenting all apps and functionalities. To consolidate the coding schema, a codebook was created to define each documented functionality. To confirm inter-rater reliability, a second review involved authors cross-checking a different section

**Table 1. COVID-19 apps features and a description of intended functionality.**

| Listed Features | Functionality Description |
|---|---|
| COVID-19 Info | Information of COVID-19 spread and/or statistics in the country |
| Route Tracking/Journal | Personal location log for contact tracing |
| Symptom Tracker | To track and log symptoms |
| Self-Assessment | Individual assessment of symptoms to determine health status |
| Quarantine Information/ Enforcement | Information about quarantining. Enforces quarantine requirements by tracking individual's location |
| Medical Reporting | Reporting symptoms to health officials and doctors through an app |
| Chatbot | Virtual assistant allows individuals to type questions into a chat box and receive an answer |
| Contact Health Officials/Helpline | Access to health officials/helpline/hotline for COVID-19 information |
| News | Government news about COVID-19 and official regulations |
| Test scheduling/results | Schedule COVID-19 test and see results |
| Health Advice/What to do | Information/health advice on how to treat COVID-19 and steps to take if diagnosed. |
| FAQ | Frequently asked questions |
| Direction/Map to Hospitals | Gives directions to nearby hospitals |
| Health status/vaccination status | Viewable health and vaccination status of an individual—digital card to show vaccination status |
| Vaccination schedule (manage/ reminder) | Vaccination scheduling management and reminder |
| Vaccination Certificate | Certificate of proof for vaccination |
| QR code of health status | Scannable individual QR code to show one's health status |
| QR code for public places | Scan QR code at public places to track location |
| Calendar | General calendar |
| General health | General health (not COVID-19 related) information |
| Border crossing registration | App provided forms to register for crossing regional borders |
| Location risk assessment | Assesses the risk of COVID-19 in a public place |
| Health status color system | Assigns color according to an individual's health status |
| Vaccination Info | Information on the COVID-19 vaccine |
| COVID-19 surveillance dashboard | Dashboard of latest COVID-19 information and numbers/statistics |
| Information for trade/business | Specific COVID-19 instructions or updates related to markets and specific businesses |
| Special needs access | App access to the deaf and blind via video instructions and voice details |

of the country list than they had previously reviewed. All discrepancies were then addressed in a meeting between the three researchers and missing eligible apps were added to apps list. See Table 1 for the resulting codes and definitions.

## Results

The results describe characteristics and functionality of 224 COVID-19 mobile apps available globally. The data were drawn from 195 countries, including the United States and each of its 50 states.

### Voluntary vs mandated

Of the 224 COVID-19 apps, 150 apps (67·0%) were available for voluntary use where the user may choose to opt-in or out at their convenience. Conversely, 15 apps (6·7%) mandated user adoption and the remaining 59 apps were unclassified (26·3%). Mandates were enforced by

either the government directly (n = 7) where there were fines and consequences for not using the apps, or by the institution (e.g., school, airports, etc.) (n = 5), or by workplace (n = 3).

### EN vs Non EN

Of the total, 128 apps (57·1%) were EN apps and 96 (42·9%) were non EN apps with 75 of the EN apps (58·6%) built with the Google Apple Exposure Notification API (GAEN). Within the 75 GAEN apps, 15 apps (10·4% of all EN apps) used the Exposure Notification Express (ENX) system. The remaining 53 apps (41·4%) used an API other than GAEN (see Fig 2).

### EN apps functions

The primary function of the 128 Exposure Notification Apps is to support communication of exposure to SARS-COV-2 virus. Most apps included additional features including the availability of COVID-19 information (n = 22, 17·2% of apps), self-assessment guidance (n = 21, 16·4% of apps), health advice (n = 20, 15·6% of apps), test scheduling and test results (n = 13, 10·2% of apps) and contact information for health officials (n = 12, 9·4%) (see Fig 3 below).

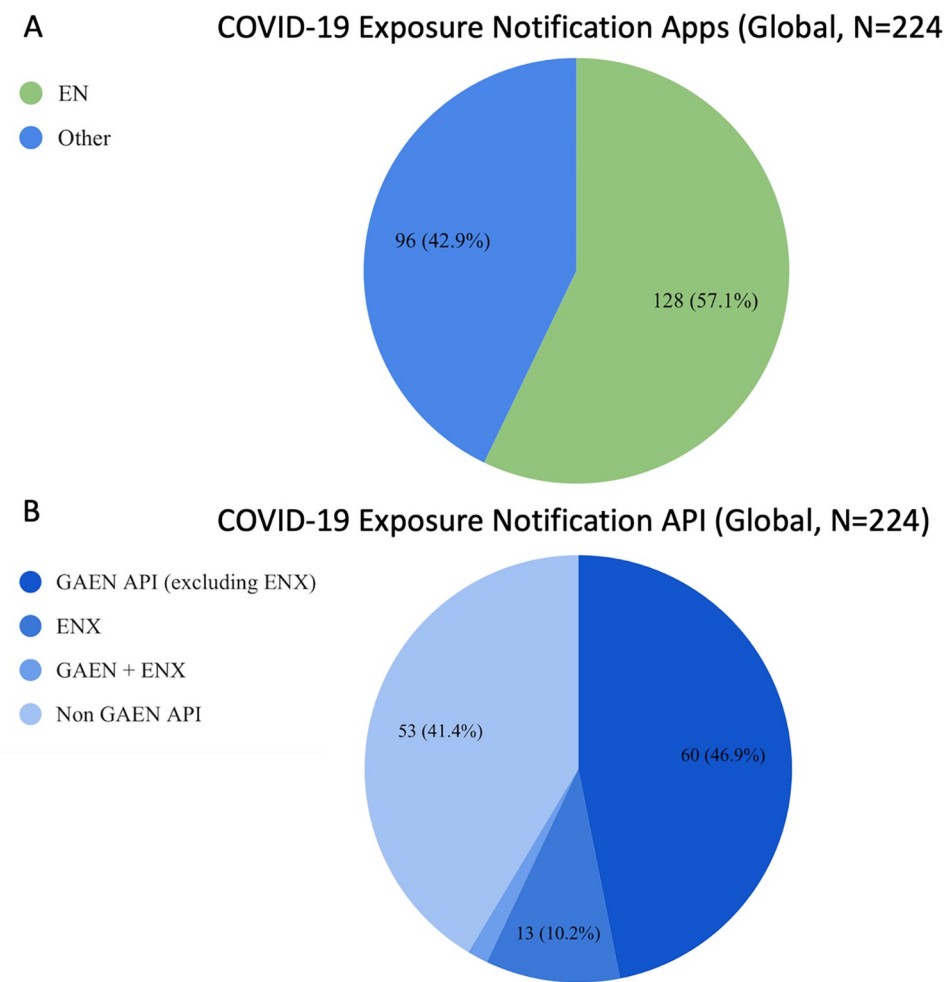

**Fig 2. EN App Features: 2A compares the numbers of exposure notification and non-exposure notification apps globally, and 2B presents the type of API technology used for exposure notification apps.**

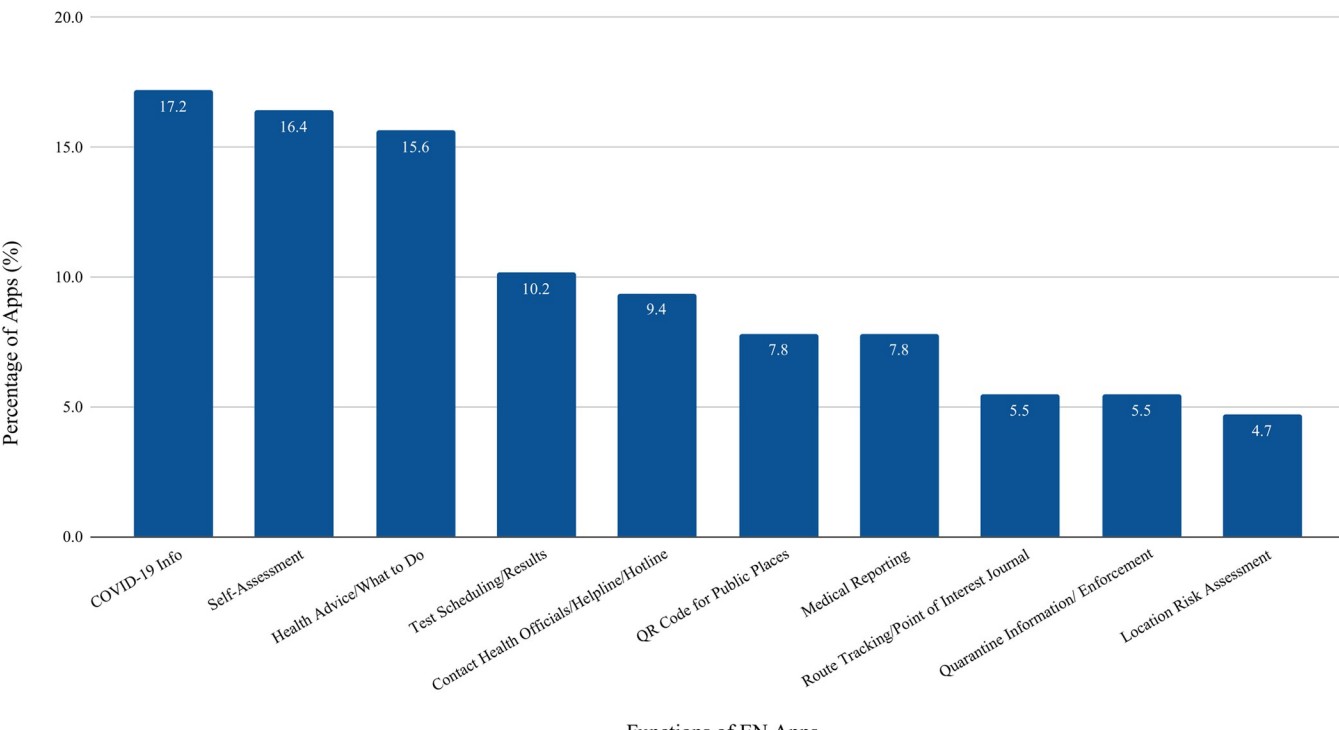

**Fig 3. Exposure Notification Functions: Fig 3 shows the most prevalent functions present across 128 exposure notification apps.**

## Non EN apps features

The features in the 96 non EN apps are similar to the EN apps except they do not include exposure notification. The top 5 features found in non EN apps were self-assessment guidance (n = 34, 35·4% of apps), COVID-19 information and statistics (n = 31, 32·3% of apps), health advice (n = 28, 29·2% of apps), quarantine information and enforcement (n = 21, 21·9% of apps), and contact tracing (n = 17, 17·7%) (see Fig 4 below). Contact tracing in non EN apps differed from contact tracing in EN apps. In the non EN Apps, the app offered an option to report to PHAs if the individual had tested positive for COVID-19. The PHA would then carry out the contact tracing process, instead of it occurring digitally like in the EN apps.

## ENX apps features

As noted previously, Exposure Notification Express (ENX) applications use the GAEN foundational platform but, can be tailored to fit the needs of a specific state or organization, as such, the features are like EN. For example, Hawaii's AlohaSafe Alert, Nevada's Covid Trace Nevada, and Virginia's COVIDWISE apps maintain unique app features because they deployed an original GAEN exposure notification application and then later adopted the ENX turnkey solution, which maintained their original app features.

There were 15 ENX applications identified with all 15 being used in the US. Fig 5 illustrates the percentage of apps containing the specified function. The top 5 features found in ENX applications include Upload Your COVID-19 Test Results (n = 15, 100·0% of apps), Anonymously Alert Others of Exposure (n = 15, 100·0% of apps), Review Your COVID-19 Exposure

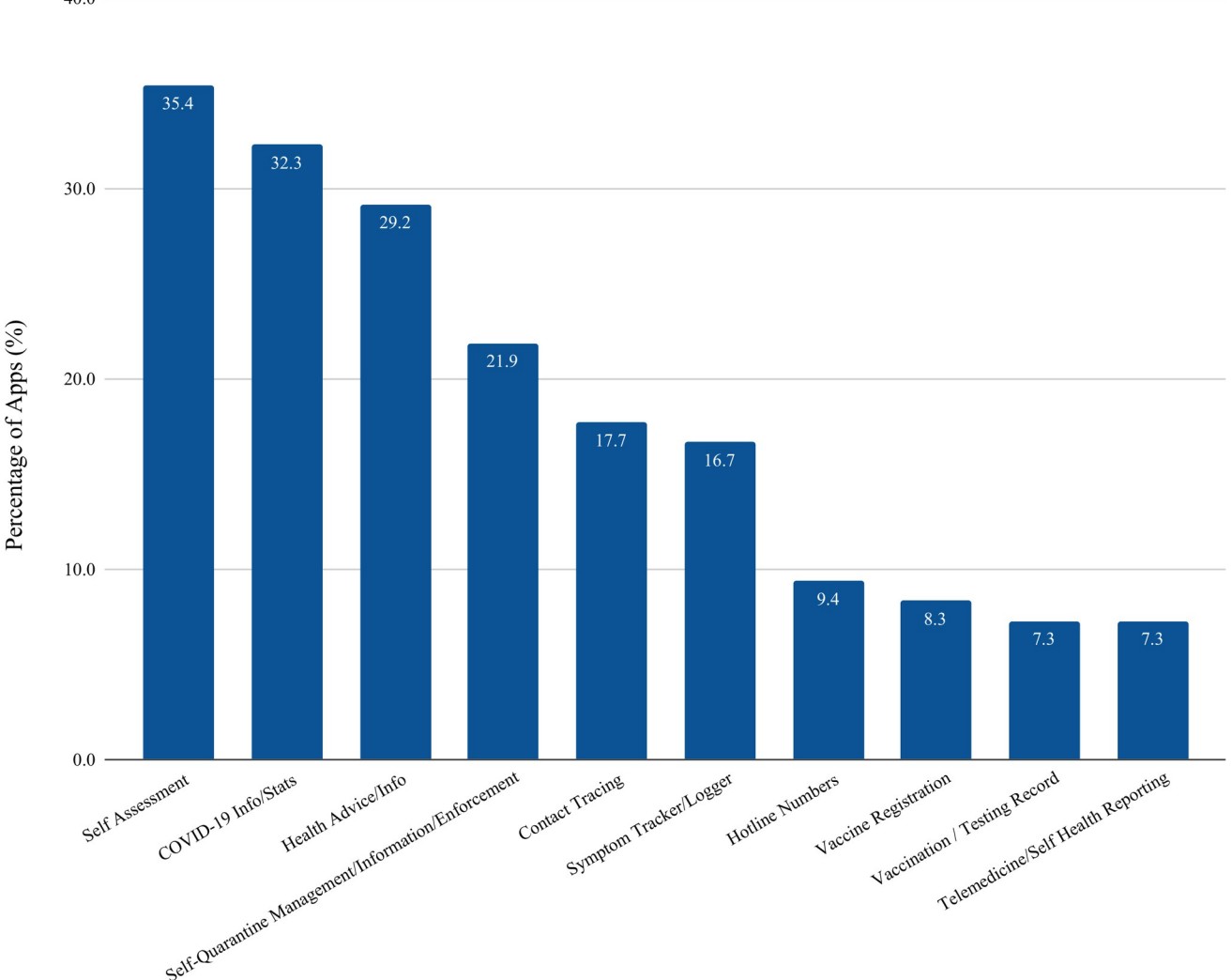

**Fig 4. Non-Exposure Notification Functions: Fig 4 depicts the most prevalent functions present across the 96 non- exposure notification apps.**

Status (n = 15, 100·0% of apps), Learn about EN (n = 13, 14·3% of apps), and Share the Word/ Invite Others (n = 13, 14·3% of apps).

## GAEN and Non-GAEN apps features

Fig 6 depicts the prevalence of functions in apps using the GAEN API (n = 75). The most common features include: COVID-19 Info (n = 13, 17·3% of apps), Self-Assessment (n = 12, 16·0% of apps), Health Advice/What to Do (n = 11, 14·7% of apps), Contact Health Officials/Help-line, (n = 6, 8·0% of apps), Medical Reporting (n = 6, 8·0% of apps), and Test Scheduling/ Results (n = 6, 8·0% of apps).

Apps not using the GAEN API (non-GAEN) make up 41.4% (n = 53) of COVID-19 Exposure Notification apps. Fig 7 identifies the most prevalent features within the EN apps that use

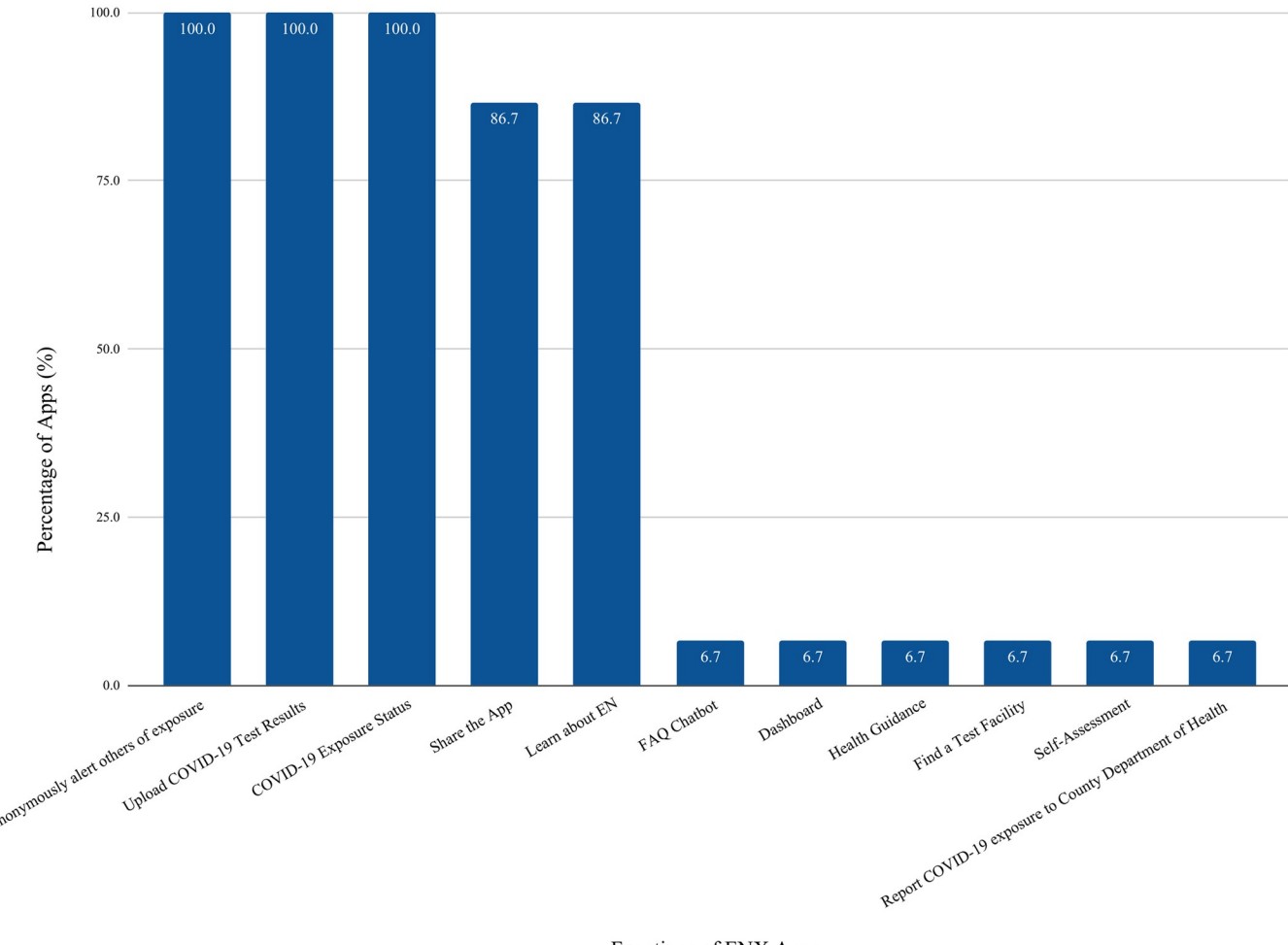

**Fig 5. ENX Apps Functions: Fig 5 illustrates the most prevalent functions present across 15 ENX apps.**

an API other than GAEN. The following are the five most prevalent functions found in non-GAEN apps: COVID-19 information (n = 9, 17.0% of apps), Health Advice/What to Do (n = 9, 17.0% of apps), Self-Assessment (n = 8, 15.1% of apps), QR Code for public places (n = 8, 15.1% of apps), and Contact Health Officials (n = 6, 11.3% of apps). Compared with GAEN apps, they share the same top three most prevalent functions: COVID-19 information, Self-Assessment, and Health Advice/What to Do. Regardless of whether the apps used GAEN or non GAEN API, EN apps share many of the same core features. Non GAEN apps had additional features, including: QR codes for public places (15.1% of non GAEN apps), and quarantine enforcement/information (11.3% of non GAEN apps), which were not present in GAEN apps. The difference in functions can be attributed to the different main purpose of GAEN based apps versus non GAEN apps and the needs and features prioritized by where they were adopted.

## Global digital EN landscape

Fig 8 shows a color-coded geographic map of all the countries/regions using COVID-19 Exposure Notification Apps by API. The notation (2+) indicates that more than 2 of that app's type

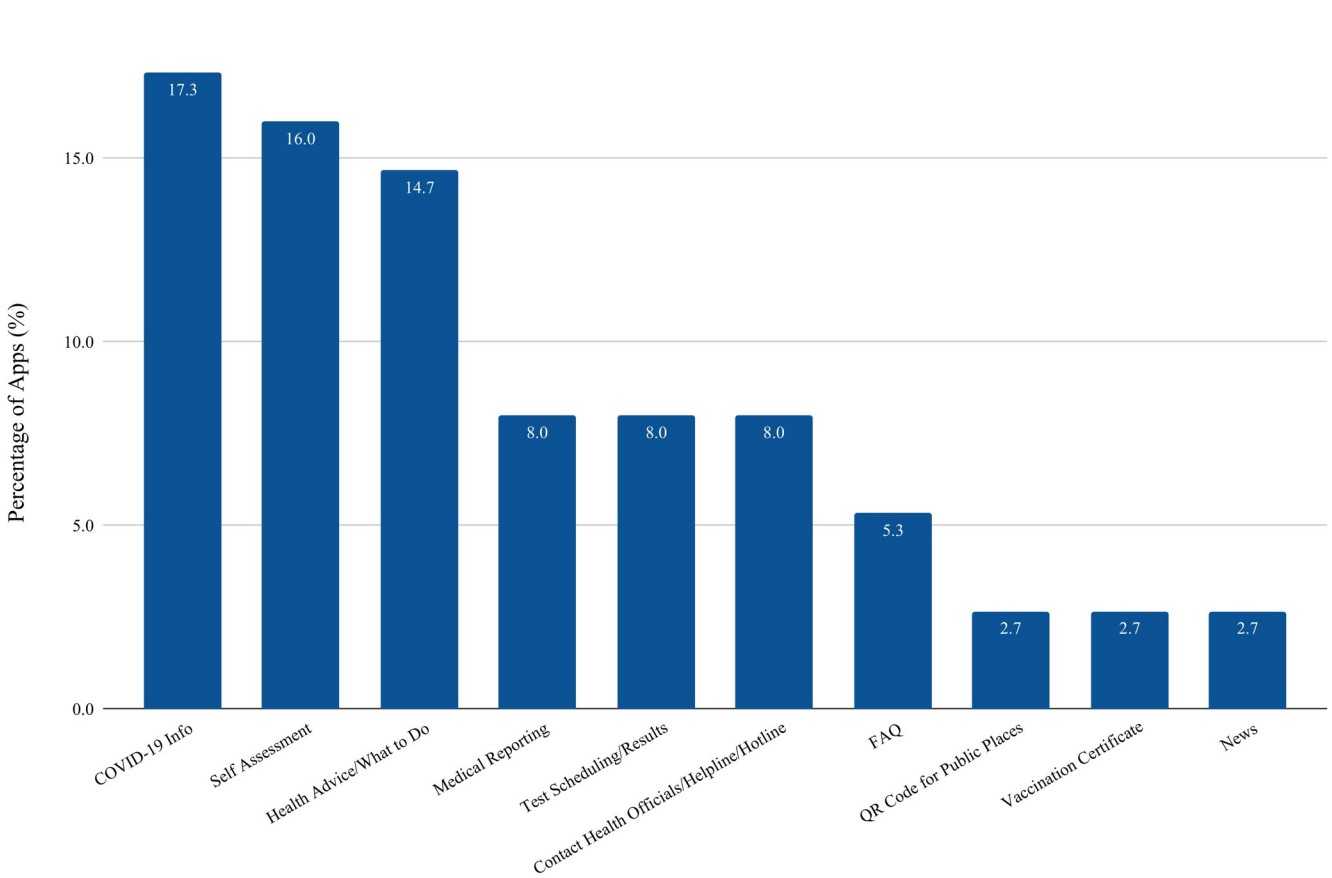

**Fig 6. Functions in GAEN apps per total count of functions found in GAEN apps: describes functions in GAEN Apps per Total Number of GAEN Apps: this table describes the functions found in GAEN apps and how many of the GAEN apps found in the search had those functions present in them.**

have been deployed within the region. EN apps not using the GAEN API were documented as a "non GAEN" app. The documentation "GAEN + non GAEN" means that the region had deployed both a GAEN app and a non GAEN app. ENX (GAEN) means that the region deployed an Exposure Notifications Express app, which also uses the GAEN API.

Designers of the apps and governments seemed to be aware of privacy being a concern based on the way some apps were designed and advertised with users' privacy in mind. Our findings show that 56 Apps, 25% of the total Apps, had some sort of privacy messaging or claimed to value privacy; whereas 44% of Exposure Notification apps included a form of privacy first messaging. For example, in Jordan's description for its EN app, Aman, the description states:

> "Your privacy matters; it is as important as your health. Therefore, AMAN only stores data on your phone and does not request any personal information or data that could lead to your identification or to breaching your privacy."

## Most Prevalent Functions Across EN Apps with Non GAEN API (n=53)

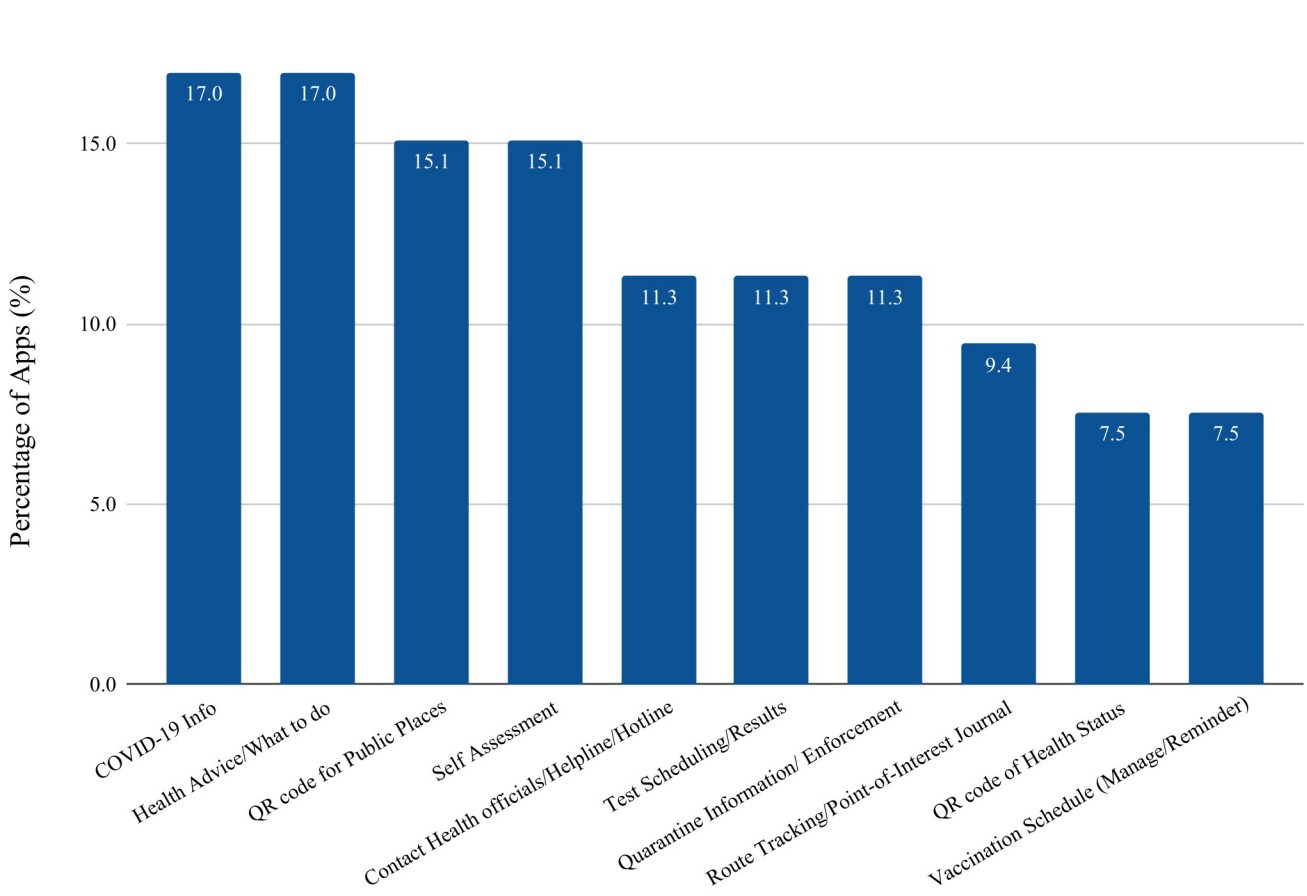

**Fig 7. Functions in Non GAEN EN apps:** describes functions in Non GAEN Apps per Total Number of Non GAEN Apps: this table describes the functions found in Non GAEN apps and how many of the Non GAEN apps found in the search had those functions present in them.

### Data management

Thirty-three of the 224 apps (14·7%) had centralized data collection where data collected by the app were managed through a centralized government server or institutional data storage system. Ninety-three (41·5%) COVID apps used a decentralized data management system, with data saved on the phone that had the app, and was not transferred or saved somewhere else.

### US vs global COVID apps

The US had a total of 33 COVID-19 Apps being used across the 50 states and Washington D. C.; however, no app was available nationally. Twenty-seven of these (81.8%) were EN apps. Each state had its own unique app and all the US apps that were made by the state and, not for a work or institution, were opt-in. The US has the highest number of COVID-19 apps compared to other countries.

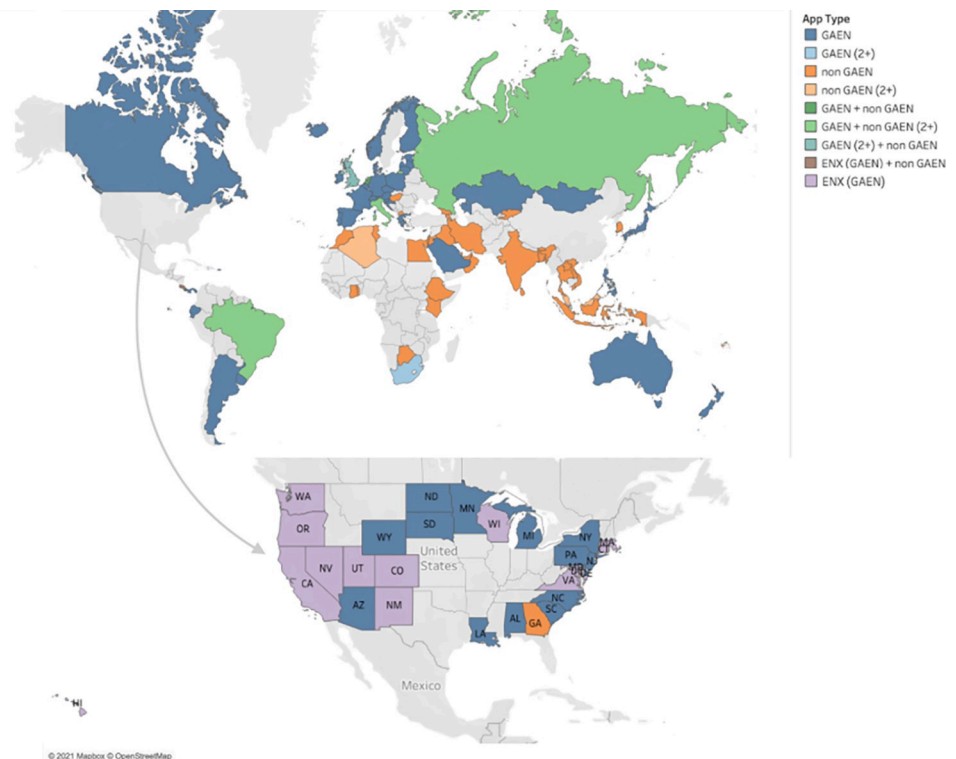

**Fig 8. Global map of regions categorized by Exposure Notification (EN) App API: The global map was created using Tableau software which uses OpenStreetmap for the basemap shapefile.** OpenStreetMap is *open data*, licensed under the Open Data Commons Open Database License (ODbL) by the OpenStreetMap Foundation (OSMF). Link to Copyright License.

## Context

While COVID-19 exposure notification apps are intended to limit the spread of infection, the existence of a COVID-19 EN app does not necessarily define the density of COVID-19 infections in the countries that used them. The countries with the lowest prevalence of COVID-19 did not necessarily have COVID-19 apps. Moreover, limited data exists regarding adoption as well as effectiveness in mitigating spread due to the recentness of COVID-19, the novelty of exposure notification technology and privacy protections.

## Discussion

The COVID-19 pandemic coincided with and spurred advances in the development and demand for novel health technologies, supporting remote monitoring, telemedicine, and digital public health interventions such as EN [13]. In this light, it is informative to compare how managing the spread of COVID differed from past pandemics. A recent study of COVID-19 technology reported the types of technologies used and adopted in digital exposure notification, including which countries adopted the various technologies and related challenges and concerns with each[9]. Our research adds to this literature by providing the first comprehensive global analysis of all COVID-19 mobile apps, their APIs, and functions provided by those apps for both Exposure Notification Apps (EN) and non-Exposure Notification Apps (non EN).

Digital contact tracing and exposure notification facilitate access, speed of information exchange, and can be scaled in a manner that is not achievable through traditional contact

tracing methods. Another benefit is the infrastructure, resources and people needed to support traditional contact tracing may be quickly overwhelmed and digital solutions may allow human resources to be more effectively applied. While digital contact tracing and exposure notification sound the same, they have some fundamental differences. Digital contact tracing in COVID-19 apps was a new feature where infected persons can use their phones to send a notification to responsible health officials or potentially exposed parties. Exposure notification technologies may use WiFi, Global Positioning System (GPS), QR Codes, or Bluetooth technology to record devices in proximity [9]. Once a person has been exposed or tests positive, they can notify the App by the click of a button. The App will then send an "exposure notification" message to the devices that have been in proximity. This transfer of proximity information can happen anonymously, when using stored Bluetooth data. Among the 36 countries that had more than one COVID App, these apps were either complementary, where some apps operated as exposure notification apps, and the other for COVID stats and updates, or they were for specific provinces and institutions within the country. Eight countries had more than one EN app, which tended to result from simple competition or specificity with a region. Some countries had both their own Exposure Notification app and a GAEN app. It would be interesting to see, as data become available, which was used most and was more efficient in controlling disease spread.

Distinguishing COVID-19 Exposure Notification (EN) apps by their API can be useful in determining what makes an app adoptable and successful in reaching its audience. Although within this study there were limitations in measuring the adoption and usage of apps, doing so would offer insight as to what makes an app attractive and achieve higher rates of adoption over other apps. The measurement of an exposure notification system's effectiveness in limiting the spread of infectious diseases, which was not conducted within this study due to various limitations, would also be of great significance for the development of future exposure notification apps.

### Privacy and human centered design

A Swiss Survey found that only 1 out of 3 citizens downloaded their EN app, SwissCovid, for which 24% of respondents reported privacy is a concern [14]. Notably, most apps were designed within a few months of COVID-19 being declared as a pandemic on March 11th, 2020 [1], demonstrating the wide acceptance of COVID-19 apps as a valuable resource. Most apps had several updates following their launch to improve and update based on COVID status and users' needs. While there was little mention of human centered design methods in COVID-19 apps, updates following product launch may have incorporated messaging to elicit trust.

### Functions

Our data reveal a global consensus that technology can significantly contribute to the mitigation of COVID-19, whether through exposure notification, general information, or any other functions noted. The three most prevalent functions in both EN and Non EN apps were COVID-19 info, self-assessment, and health advice, which demonstrates a majority COVID-19 apps are used to inform the public regardless of EN capabilities. All EN apps, as expected, had the primary function of notification; therefore, secondary functions were not likely prioritized during product development (e.g., a main function of providing COVID-19 information was 16% in EN and 33% in Non EN apps). Given applications maybe more acceptable with added features, developers may want to consider providing more app functions in future iterations.

## Limitations and future directions

Due to the fast-paced development and deployment of COVID-19 apps and the uncertainties of a global pandemic, there was limited research available about app adoption or effectiveness. While incomplete, the findings from this study provide insights into technology use in this current pandemic and can be leveraged to learn shortcomings of COVID-19 apps for future pandemics and epidemics.

Possible bias or inaccuracy may have occurred when apps found were only accessible in their respective country's native language. Google Translate was used to interpret the official app pages and the graphic images describing the function of the apps in the App Store and Google Play. In addition, personalized algorithms could influence which apps the Google Search engine pushed prioritized on the results page. Another limitation to our search is that we may not have uncovered all COVID-19 related apps as some were blocked for individuals out of the country of use, i.e., there were some countries where COVID-19 related apps showed up in the initial research, but no additional details could be found.

Within this study, apps using the GAEN API make up 56·0% of COVID-19 exposure notification apps globally. Analyzing the adoption rates and impact of apps using the GAEN API compared to apps using other systems would be beneficial for developing future technology with higher adoption and effectiveness. This study did not pursue data on adoption rates due to lack of current availability. However, future research should pursue adoption metrics to identify correlations between certain functions and ways to increase acceptance and, subsequent usage.

## Conclusions

This global landscape analysis reveals how mobile apps are being used to manage the spread of COVID-19. The review of 224 COVID-19 mobile apps identified two main type of apps: Exposure Notification apps and non-Exposure Notification apps. Exposure Notification apps have a significant potential to supplement traditional contact tracing by providing cost effective and efficient methods for stopping viral spread. Apps that do not serve to notify of exposure provide COVID related information and support. Research is needed to understand barriers and facilitators to adoption and how the variety of COVID-19 app features play into public acceptance and use.

## Supporting information

**S1 Table. COVID-19 Exposure Notification Apps: describes all of the Covid-19: exposure notifications app that came up in the search and which ones are Google and Apple Exposure Notifications (GAEN) apps and which ones are not GAEN based.**
(TIFF)

**S2 Table. COVID-19 Non Exposure Notification Apps: describes all the non-exposure notifications app that came up in the search as covid related apps but were not covid exposure apps.**
(TIFF)

**S3 Table. Features in EN Apps per Total Number of EN Apps: this table describes the features found in EN apps and how many of the EN apps found in the search had those features present in them.**
(TIFF)

**S4 Table. Non EN App Features per Total Number of non EN Apps: describes features in Non EN Apps per Total Number of Non EN Apps: this table describes the features found in Non EN apps and how many of the Non EN apps found in the search had those features present in them.**
(TIFF)

**S5 Table. GAEN App Functions per Total Number of GAEN Apps: describes features in GAEN Apps per Total Number of GAEN Apps: this table describes the features found in GAEN apps and how many of the GAEN apps found in the search had those features present in them.**
(TIFF)

**S6 Table. Functions in non GAEN EN Apps per Total Number of non GAEN EN Apps: describes features in Non GAEN Apps per Total Number of Non GAEN Apps: this table describes the features found in Non GAEN apps and how many of the Non GAEN apps found in the search had those features present in them.**
(TIFF)

## Author Contributions

**Conceptualization:** Camille Nebeker, Mohsen Malekinejad, Eliah Aronoff-Spencer.

**Data curation:** Daniah Kareem, Aidan Yong, Eliah Aronoff-Spencer.

**Formal analysis:** Camille Nebeker, Daniah Kareem, Aidan Yong, Rachel Kunowski, Eliah Aronoff-Spencer.

**Funding acquisition:** Eliah Aronoff-Spencer.

**Investigation:** Camille Nebeker, Daniah Kareem, Aidan Yong, Rachel Kunowski, Eliah Aronoff-Spencer.

**Methodology:** Camille Nebeker, Eliah Aronoff-Spencer.

**Project administration:** Camille Nebeker, Daniah Kareem, Rachel Kunowski, Mohsen Malekinejad, Eliah Aronoff-Spencer.

**Resources:** Eliah Aronoff-Spencer.

**Supervision:** Camille Nebeker, Eliah Aronoff-Spencer.

**Validation:** Aidan Yong.

**Visualization:** Daniah Kareem, Aidan Yong, Rachel Kunowski.

**Writing – original draft:** Camille Nebeker, Daniah Kareem, Eliah Aronoff-Spencer.

**Writing – review & editing:** Camille Nebeker, Daniah Kareem, Aidan Yong, Rachel Kunowski, Mohsen Malekinejad, Eliah Aronoff-Spencer.

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
